# Symbolic Opportunistic Policy Iteration for Factored-Action MDPs

**Aswin Raghavan**[a] **Roni Khardon**[b] **Alan Fern**[a] **Prasad Tadepalli**[a]
[a] School of EECS, Oregon State University, Corvallis, OR, USA
{nadamuna,afern,tadepall}@eecs.orst.edu
[b] Department of Computer Science, Tufts University, Medford, MA, USA
roni@cs.tufts.edu

## Abstract

This paper addresses the scalability of symbolic planning under uncertainty with factored states and actions. Our first contribution is a symbolic implementation of Modified Policy Iteration (MPI) for factored actions that views policy evaluation as policy-constrained value iteration (VI). Unfortunately, a naïve approach to enforce policy constraints can lead to large memory requirements, sometimes making symbolic MPI worse than VI. We address this through our second and main contribution, symbolic Opportunistic Policy Iteration (OPI), which is a novel convergent algorithm lying between VI and MPI, that applies policy constraints if it does not increase the size of the value function representation, and otherwise performs VI backups. We also give a memory bounded version of this algorithm allowing a space-time tradeoff. Empirical results show significantly improved scalability over state-of-the-art symbolic planners.

## 1   Introduction

We study symbolic dynamic programming (SDP) for Markov Decision Processes (MDPs) with exponentially large factored state and action spaces. Most prior SDP work has focused on exact [1] and approximate [2, 3] solutions to MDPs with factored states, assuming just a handful of atomic actions. In contrast to this, many applications are most naturally modeled as having factored actions described in terms of multiple *action variables*, which yields an exponential number of joint actions. This occurs, e.g., when controlling multiple actuators in parallel, such as in robotics, traffic control, and real-time strategy games. In recent work [4] we have extended SDP to factored actions by giving a symbolic VI algorithm that explicitly reasons about action variables. The key bottleneck of that approach is the space and time complexity of computing symbolic Bellman backups, which requires reasoning about all actions at all states *simultaneously*. This paper is motivated by addressing this bottleneck via the introduction of alternative and potentially much cheaper backups.

We start by considering Modified Policy Iteration (MPI) [5], which adds a few policy evaluation steps between consecutive Bellman backups. MPI is attractive for factored-action spaces because policy evaluation does not require reasoning about all actions at all states, but rather only about the current policy's action at each state. Existing work on symbolic MPI [6] assumes a small atomic action space and does not scale to factored actions. Our first contribution (Section 3) is a new algorithm, Factored Action MPI (FA-MPI), that conducts exact policy evaluation steps by treating the policy as a constraint on normal Bellman backups.

While FA-MPI is shown to improve scalability compared to VI in some cases, we observed that in practice the strict enforcement of the policy constraint can cause the representation of value functions to become too large and dominate run time. Our second and main contribution (Section 4) is to overcome this issue using a new backup operator that lies between policy evaluation and a Bellman

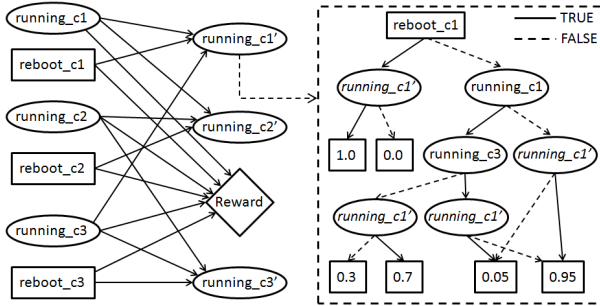

Figure 1: Example of a DBN MDP with factored actions.

backup and hence is guaranteed to converge. This new algorithm, Opportunistic Policy Iteration (OPI), constrains a select subset of the actions in a way that guarantees that there is no growth in the representation of the value function. We also give a memory-bounded version of the above algorithm (Section 5). Our empirical results (Section 6) show that these algorithms are significantly more scalable than FA-MPI and other state-of-the-art algorithms.

## 2    MDPs with Factored State and Action Spaces

In a factored MDP $M$, the state space $\mathbb{S}$ and action space $\mathbb{A}$ are specified by finite sets of binary variables $\mathbf{X} = (X_1, \ldots, X_l)$ and $\mathbf{A} = (A_1, \ldots, A_m)$ respectively, so that $|\mathbb{S}| = 2^l$ and $|\mathbb{A}| = 2^m$. For emphasis we refer to such MDPs as factored-action MDPs (FA-MDPs). The transition function $T$ and reward function $R$ are specified compactly using a Dynamic Bayesian Network (DBN). The DBN model consists of a two–time-step graphical model that shows, for each next state variable $X'$ and the immediate reward, the set of current state and action variables, denoted by parents$(X')$. Further, following [1], the conditional probability functions are represented by algebraic decision diagrams (ADDs) [7], which represent real-valued functions of boolean variables as a Directed Acyclic Graph (DAG) (i.e., an ADD maps assignments to $n$ boolean variables to real values). We let $P^{X'_i}$ denote the ADD representing the conditional probability table for variable $X'_i$.

For example, Figure 1 shows a DBN for the SysAdmin domain (Section 6.1). The DBN encodes that the computers c1, c2 and c3 are arranged in a directed ring so that the running status of each is influenced by its reboot action and the status of its predecessor. The right part of Figure 1 shows the ADD representing the dynamics for the state variable running_c1. The variable running_c1' represents the truth value of running_c1 in the next state. The ADD shows that running_c1 becomes true if it is rebooted, and otherwise the next state depends on the status of the neighbors. When not rebooted, c1 fails w.p. 0.3 if its neighboring computer c3 has also failed, and w.p. 0.05 otherwise. When not rebooted, a failed computer becomes operational w.p. 0.05.

ADDs support binary operations over the functions they represent ($F \ op \ G = H$ if and only if $\forall x, F(x) \ op \ G(x) = H(x)$) and marginalization operators (e.g., marginalize $x$ via maximization in $G(y) = \max_{\mathbf{x}} F(x, y)$ and through sum in $G(y) = \sum_{\mathbf{x}} F(x, y)$ ). Operations between diagrams will be represented using the usual symbols $+, \times, \max$ etc., and the distinction between scalar operations and operations over functions should be clear from context. Importantly, these operations are carried out symbolically and scale polynomially in the size of the ADD rather than the potentially exponentially larger tabular representation of the function. ADD operations assume a total ordering $O$ on the variables and impose that ordering in the DAG structure (interior nodes) of any ADD. SDP uses the compact MDP model to derive compact value functions by iterating symbolic Bellman backups that avoid enumerating all states. It has the advantage that the value function is exact while often being much more compact than explicit tables. Early SDP approaches such as SPUDD [1] only represented the structure in the state variables and enumerate over actions, so that space and time is at least linearly related to the number of actions, and hence exponential in $m$.

In recent work, we extended SDP to factored action spaces by computing Bellman backups using an algorithm called Factored Action Regression (FAR) [4]. This is done by implementing the following equations using ADD operations over a representation like Figure 1. Let $\mathcal{T}^Q(V)$ denote the backup

operator that computes the next iterate of the Q-value function starting with value function $V$,

$$\mathcal{T}^Q(V) = R + \gamma \sum_{X_1'} P^{X_1'} \ldots \sum_{X_l'} P^{X_l'} \times primed(V) \qquad (1)$$

then $\mathcal{T}(V) = \max_{A_1} \ldots \max_{A_m} \mathcal{T}^Q(V)$ gives the next iterate of the value function. Repeating this process we get the VI algorithm. Here $primed(V)$ swaps the state variables $\mathbf{X}$ in the diagram $V$ with next state variables $\mathbf{X}'$ (c.f. DBN representation for next state variables). Equation 1 should be read right to left as follows: each probability diagram $P^{X_i'}$ assigns a probability to $X_i'$ from assignments to $Parents(X_i') \subseteq (\mathbf{X}, \mathbf{A})$, introducing the variables $Parents(X_i')$ into the value function. The $\sum$ marginalization eliminates the variable $X_i'$. We arrive at the Q-function that maps variable assignments $\subseteq (\mathbf{X}, \mathbf{A})$ to real values. Written in this way, where the domain dynamics are explicitly expressed in terms of actions variables and where $\max_{\mathbf{A}} = \max_{A_1, \ldots, A_m}$ is a symbolic marginalization operation over action variables, we get the Factored Action Regression (FAR) algorithm [4]. In the following, we use $T()$ to denote a Bellman-like backup where superscript $T^Q()$ denotes that that actions are not maximized out so the output is a function of state and actions, and subscript as in $T_\pi()$ defined below denotes that the update is restricted to the actions in $\pi$. Similarly $T_\pi^Q()$ restricts to a (possibly partial) policy $\pi$ and does not maximize over the unspecified action choice.

In this work we will build on Modified Policy Iteration (MPI), which generalizes value iteration and policy iteration, by interleaving $k$ policy evaluation steps between successive Bellman backups [5]. Here a policy evaluation step corresponds to iterating exact policy backups, denoted by $T_\pi$ where the action is prescribed by the policy $\pi$ in each state. MPI has the potential to speed up convergence over VI because, at least for flat action spaces, policy evaluation is considerably cheaper than full Bellman backups. In addition, when $k > 0$, one might hope for larger jumps in policy improvement because the greedy action in $\mathcal{T}$ is based on a more accurate estimate of the value of the policy.

Interestingly, the first approach to symbolic planning in MDPs was a version of MPI for factored states called Structured Policy Iteration (SPI), which was [6] later adapted to relational problems [8]. SPI represents the policy as a decision tree with state-variables labeling interior nodes and a concrete action as a leaf node. The policy backup uses the graphical form of the policy. In each such backup, for each leaf node (policy action) $\mathbf{a}$ in the policy tree, its Q-function $Q_{\mathbf{a}}$ is computed and attached to the leaf. Although SPI leverages the factored state representation, it represents the policy in terms of concrete joint actions, which fails to capture the structure among the action variables in FA-MDPs. In addition, in factored actions spaces this requires an explicit calculation of $Q$ functions for all joint actions. Finally, the space required for policy backup can be prohibitive because each $Q$-function $Q^{\mathbf{a}}$ is joined to each leaf of the policy. SPI goes to great lengths in order to enforce a policy backup which, intuitively, ought to be much easier to compute than a Bellman backup. In fact, we are not aware of any implementations of this algorithm that scales well for FA-MDPs or even for factored state spaces. The next section provides an alternative algorithm.

## 3 Factored Action MPI (FA-MPI)

In this section, we introduce Factored Action MPI (FA-MPI), which uses a novel form of policy backup. Pseudocode is given in Figure 2. Each iteration of the outer while loop starts with one full Bellman backup using Equation 1, i.e., policy improvement. The inner loop performs $k$ steps of policy backups using a new algorithm described below that avoids enumerating all actions.

We represent the policy using a Binary Decision Diagram (BDD) with state and action variables where a leaf value of 1 denotes any combination of action variables that is the policy action, and a leaf value of $-\infty$ indicates otherwise. Using this representation, we perform policy backups using $T_\pi^Q(V)$ given in Equation 2 below followed by a max over the actions in the resulting diagram. In this equation, the diagram resulting from the product $\pi \times primed(V)$ sets the value of all off-policy state-actions to $-\infty$, before computing any value for them[1] and this ensures correctness of the update as indicated by the next proposition.

$$T_\pi^Q(V) = \left[ R + \gamma \sum_{X_1'} P^{X_1'} \ldots \sum_{X_l'} P^{X_l'} \times (\pi \times primed(V)) \right] \qquad (2)$$

**Algorithm 3.1:** FA-MPI/OPI(k)

$V^0 \leftarrow 0, i \leftarrow 0$
$(V_0^{i+1}, \pi^{i+1}) \leftarrow \max_{\mathbf{A}} T^Q(V^i)$
**while** $||V_0^{i+1} - V^i|| > \epsilon$

$\mathbf{do} \begin{cases} \mathbf{for}\ j \leftarrow 1\ \mathbf{to}\ k \\ \quad \mathbf{do} \begin{cases} \text{For Algorithm FA-MPI :} \\ \quad V_j^{i+1} \leftarrow \max_{\mathbf{A}} T_{\pi^{i+1}}^Q(V_{j-1}^{i+1}) \\ \text{For Algorithm OPI :} \\ \quad V_j^{i+1} \leftarrow \max_{\mathbf{A}} \hat{T}_{\pi^{i+1}}^Q(V_{j-1}^{i+1}) \end{cases} \\ V^{i+1} \leftarrow V_{\mathrm{k}}^{i+1} \\ i \leftarrow i + 1 \\ (V_0^{i+1}, \pi^{i+1}) \leftarrow \max_{\mathbf{A}} T^Q(V^i) \end{cases}$

**return** $(\pi^{i+1})$.

Figure 2: Factored Action MPI and OPI.

**Algorithm 3.2:** $\mathcal{P}(\mathrm{D}, \pi)$

d $\leftarrow$ variable at the root node of D
c $\leftarrow$ variable at root node of $\pi$
**if** d occurs after c in ordering
  **then** $\mathcal{P}(\mathrm{D}, \max(\pi_T, \pi_F))$
  **else if** d = c
  **then** $ADD(\mathrm{d}, \mathcal{P}(\mathrm{D}_T, \pi_T), \mathcal{P}(\mathrm{D}_F, \pi_F))$
  **else if** d occurs before c in ordering
  **then** $ADD(\mathrm{d}, \mathcal{P}(\mathrm{D}_T, \pi), \mathcal{P}(\mathrm{D}_F, \pi))$
  **else if** $\pi = -\infty$ **return** $(-\infty)$
  **else** D

Figure 3: Pruning procedure for an ADD. Subscripts $T$ and $F$ denote the true and false child respectively.

**Proposition 1.** *FA-MPI computes exact policy backups i.e.* $\max_{\mathbf{A}} \mathcal{T}_\pi^Q = \mathcal{T}_\pi$.

The proof uses the fact that $(s, a)$ pairs that do not agree with the policy get a value $-\infty$ via the constraints and therefore do not affect the maximum. While FA-MPI can lead to improvements over VI (i.e. FAR), like SPI, FA-MPI can lead to large space requirements in practice. In this case, the bottleneck is the ADD product $\pi \times primed(V)$, which can be exponentially larger than $primed(V)$ in the worst case. The next section shows how to approximate the backup in Equation 2 while ensuring no growth in the size of the ADD.

# 4 Opportunistic Policy Iteration (OPI)

Here we describe Opportunistic Policy Iteration (OPI), which addresses the shortcomings of FA-MPI. As seen in Figure 2, OPI is identical to FA-MPI except that it uses an alternative, more conservative policy backup. The sequence of policies generated by FA-MPI (and MPI) may not all have compactly representable ADDs. Fortunately, finding the optimal value function may not require representing the values of the intermediate policies exactly. The key idea in OPI is to enforce the policy constraint *opportunistically*, i.e. only when they do not increase the size of the value function representation.

In an exponential action space, we can sometimes expect a Bellman backup to be a coarser partitioning of state variables than the value function of a given policy (e.g. two states that have the same value under the optimal action have different values under the policy action). In this case enforcing the policy constraint via $T_\pi^Q(V)$ is actually harmful in terms of the size of the representation. OPI is motivated by retaining the coarseness of Bellman backups in some states, and otherwise enforcing the policy constraint. The OPI backup is sensitive to the size of the value ADD so that it is guaranteed to be smaller than the results of both Bellman backup and policy backup.

First we describe the symbolic implementation of OPI . The trade-off between policy evaluation and policy improvement is made via a pruning procedure (pseudo-code in Figure 3). This procedure assigns a value of $-\infty$ to only those paths in a value function ADD that violate the policy constraint $\pi$. The interesting case is when the root variable of $\pi$ is ordered below the root of $D$ (and thus does not appear in $D$) so that the only way to violate the constraint is to violate both true and false branches. We therefore recurse $D$ with the diagram $\max\{\pi_T, \pi_F\}$.

**Example 1.** *The pruning procedure is illustrated in Figure 4. Here the input function $D$ does not contain the root variable $X$ of the constraint, and the max under $X$ is also shown. The result of pruning $\mathcal{P}(D, \pi)$ is no more complex than $D$, whereas the product $D \times \pi$ is more complex.*

Clearly, the pruning procedure is not sound for ADDs because there may be paths that violate the policy, but are not explicitly represented in the input function $D$. In order to understand the result of $\mathcal{P}$, let $p$ be a path from a root to a leaf in an ADD. The path $p$ induces a partial assignment to the

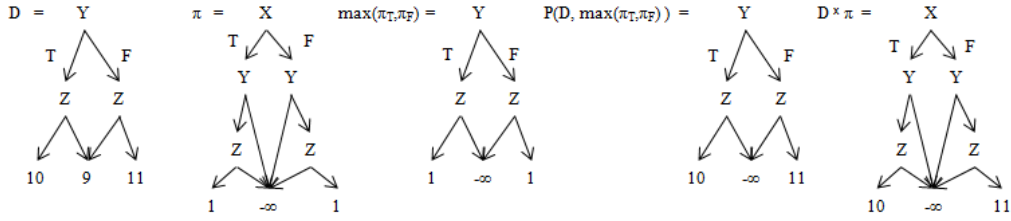

Figure 4: An example for pruning. $D$ and $\pi$ denote the given function and constraint respectively. The result of pruning is no larger than $D$, as opposed to multiplication. $T$ (true) and $F$ (false) branches are denoted by the left and the right child respectively.

variables in the diagram. Let $\mathcal{E}(p)$ be the set of all extensions of this partial assignment to complete assignments to all variables. As established in the following proposition, a path is pruned if none of its extensions satisfies the constraint.

**Proposition 2.** *Let* $G = \mathcal{P}(D, \pi)$ *where leaves in* $D$ *do not have the value* $-\infty$. *Then for all paths* $p$ *in* $G$ *we have:*
1. *$p$ leads to $-\infty$ in $G$ iff $\forall y \in \mathcal{E}(p), \pi(y) = -\infty$.*
2. *$p$ does not lead to $-\infty$ in $G$ iff $\forall y \in \mathcal{E}(p), G(y) = D(y)$.*
3. *The size of the ADD $G$ is smaller or equal to the size of $D$.*

The proof (omitted due to space constraints) uses structural induction on $D$ and $\pi$. The novel backup introduced in OPI interleaves the application of pruning with the summation steps so as to prune the diagram as early as possible. Let $\mathcal{P}_\pi(D)$ be shorthand for $\mathcal{P}(D, \pi)$. The backup used by OPI, which is shown in Figure 2 is

$$\hat{T}_\pi^Q(V) = \mathcal{P}_\pi \left[ \mathcal{P}_\pi(R) + \gamma \mathcal{P}_\pi(\sum_{X_1'} P^{X_1'} \ldots \mathcal{P}_\pi(\sum_{X_l'} P^{X_l'} \times primed(V))))) \right] \tag{3}$$

Using the properties of $\mathcal{P}$ we can show that $\hat{T}_\pi^Q(V)$ overestimates the true backup of a policy, but is still bounded by the true value function.

**Theorem 1.** *The policy backup used by OPI is bounded between the full Bellman backup and the true policy backup, i.e.* $\mathcal{T}_\pi \leq \max_{\mathbf{A}} \hat{\mathcal{T}}_\pi^Q \leq \mathcal{T}$.

Since none of the value functions generated by OPI overestimate the optimal value function, it follows that both OPI and FA-MPI converge to the optimal policy under the same conditions as MPI [5]. However, the sequence of value functions/policies generated by OPI are in general different from and potentially more compact than those generated by FA-MPI. The relative compactness of these policies is empirically investigated in Section 6. The theorem also implies that OPI converges at least as fast as FA-MPI to the optimal policy, and may converge faster.

In terms of a flat MDP, OPI can be interpreted as sometimes picking a greedy off-policy action while evaluating a fixed policy, when the value function of the greedy policy is at least as good and more compact than that of the given policy. Thus, OPI may be viewed as asynchronous policy iteration ([9]). However, unlike traditional asynchronous PI, the policy improvement in OPI is motivated by the size of the representation, rather than any measure of the magnitude of improvement.

**Example 2.** *Consider the example in Figure 5. Suppose that $\pi$ is a policy constraint that says that the action variable $A_1$ must be true when the state variable $X_2$ is false. The backup $\mathcal{T}^Q(R)$ does not involve $X_2$ and therefore pruning does not change the diagram and $P_\pi(\mathcal{T}^Q(R)) = \mathcal{T}^Q(R)$. The max chooses $A_1 = true$ in all states, regardless of the value of $X_2$, a greedy improvement. Note that the improved policy (always set $A_1$) is more compact than $\pi$, and so is its value. In addition, $P_\pi(\mathcal{T}^Q(R))$ is coarser than $\pi \times \mathcal{T}^Q(\mathcal{R})$.*

## 5   Memory-Bounded OPI

Memory is usually a limiting factor for symbolic planning. In [4] we proposed a symbolic memory bounded (MB) VI algorithm for FA-MDPs, which we refer to below as Memory Bounded Factored

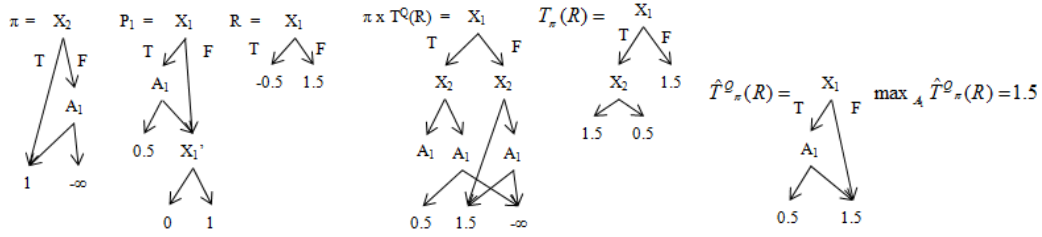

(a) A simple policy for an MDP with two state variables, $X_1$ and $X_2$, and one action variable $A_1$.

(b) Optimal policy backup in FA-MPI.

(c) OPI backup. Note the smaller size of the value function.

Figure 5: An illustration where OPI computes an incorrect but more compact value function that is is a partial policy improvement. $T$ (true) and $F$ (false) branches are denoted by the left and the right child respectively.

Action Regression (MBFAR). MBFAR generalizes SPUDD and FAR by flexibly trading off computation time for memory. The key idea is that a backup can be computed over a partially instantiated action, by fixing the value of an action variable. MBFAR computes what [10] called "Z-value functions" that are optimal value functions for partially specified actions. But in contrast to their work, where the set of partial actions are hand-coded by the designer, MBFAR is domain-independent and depends on the complexity of the value function. In terms of time to convergence, computing these subsets on the fly may lead to some overhead, but in some cases may lead to a speedup. Memory Bounded FA-MPI (MB-MPI) is a simple extension that uses MBFAR in place of FAR for the backups in Figure 2. MB-MPI is parametrized by $k$, the number of policy backups, and $M$, the maximum size (in nodes) of a Z-value function. MB-MPI generalizes MPI in that MB-MPI(k,0) is the same as SPI(k) [6] and MB-MPI(k,$\infty$) is FA-MPI(k). Also, MB-MPI(0,0) is SPUDD [1] and MB-MPI(0,$\infty$) is FAR [4]. We can also combine OPI with memory bounded backup. We will call this algorithm MB-OPI. Since both MB-MPI and OPI address space issues in FA-MPI the question is whether one dominates the other and whether their combination is useful. This is addressed in the experiments.

# 6 Experiments

In this section, we experimentally evaluate the algorithms and the contributions of different components in the algorithms.

## 6.1 Domain descriptions

The following domains were described using the Relational Dynamic Influence Diagram Language (RDDL) [11]. We ground the relational description to arrive at the MDP similar to Figure 1. In our experiments the variables in the ADDs are ordered so that $parents(X_i')$ occur above $X_i'$ and the $X_i'$s are ordered by $|parents(X_i')|$. We heuristically chose to do the expectation over state variables in the top-down way, and maximization of action variables in the bottom-up way with respect to the variable ordering.

**Inventory Control**(IC): This domain consists of $n$ independent shops each being full or empty that can be filled by a deterministic action. The total number of shops that can be filled in one time step is restricted. The rate of arrival of a customer is distributed independently and identically for all shops as $Bernoulli(p)$ with $p = 0.05$. A customer at an empty shop continues to wait with a reward of -1 until the shop is filled and gives a reward of -0.35. An instance of IC with $n$ shops and $m$ trucks has a joint state and action space of size $2^{2n}$ and $\sum_{i=0}^{m} \binom{n}{i}$ respectively.

**SysAdmin**: The "SysAdmin" domain was part of the IPC 2011 benchmark and was introduced in earlier work [12]. It consists of a network of $n$ computers connected in a given topology. Each computer is either running (reward of +1) or failed (reward of 0) so that $|S| = 2^n$, and each computer has an associated deterministic action of rebooting (with a cost of -0.75) so that $|A| = 2^n$. We restrict the number of computers that can be rebooted in one time step. Unlike the previous domain, the exogenous events are not independent of one another. A running computer that is not being

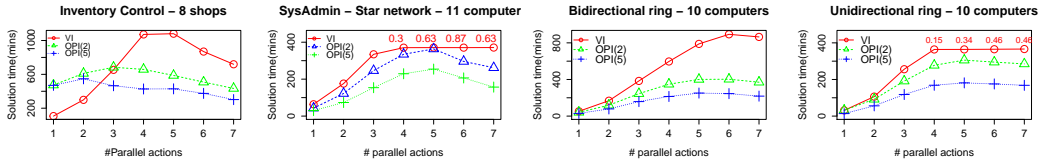

Figure 6: Impact of policy evaluation: Parallel actions vs. Time. In Star and Unidirectional networks VI was stopped at a time limit of six hours and the Bellman error is annotated.

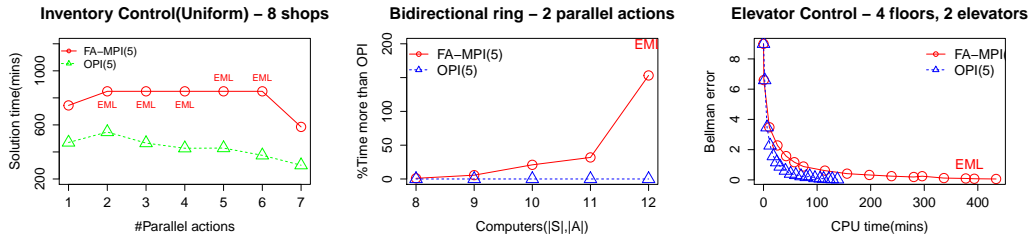

Figure 7: Impact of Pruning. EML denotes Exceeded Memory Limit and the Bellman error is denoted in parenthesis.

rebooted is running in the next state with probability $p$ proportional to the number of its running neighbors, where $p = 0.45 + 0.5\left(\frac{1+n_r}{1+n_c}\right)$, $n_r$ is the number of neighboring computers that have not failed and $n_c$ is the number of neighbors. We test this domain on three topologies of increasing difficulty, viz. a star topology, a unidirectional ring and a bidirectional ring.

**Elevator control**: We consider the problem of controlling $m$ elevators in a building with $n$ floors. A state is described as follows: for each floor, whether a person is waiting to go up or down; for each elevator, whether a person inside the elevator is going up or down, whether the elevator is at each floor, and its current direction (up or down). A person arrives at a floor $f$, independently of other floors, with a probability $Bernoulli(p_f)$, where $p_f$ is drawn from $Uniform(0.1, 0.3)$ for each floor. Each person gets into an elevator if it is at the same floor and has the same direction (up or down), and exits at the top or bottom floor based on his direction. Each person gets a reward of -1 when waiting at a floor and -1.5 if he is in an elevator that is moving in a direction opposite to his destination. There is no reward if their directions are the same. Each elevator has three actions: move up or down by one floor, or flip its direction.

## 6.2 Experimental validation

In order to evaluate scaling with respect to the action space we fix the size of the state-space and measure time to convergence (Bellman error less than 0.1 with discount factor of 0.9). Experiments were run on a single core of an Intel Core 2 Quad 2.83GHz with 4GB limit. The charts denote OPI with $k$ steps of evaluation as OPI ($k$), and MB-OPI with memory bound $M$ as MB-OPI($k, M$) (similarly FA-MPI($k$) and MB-MPI($k, M$)). In addition, we compare to symbolic value iteration:

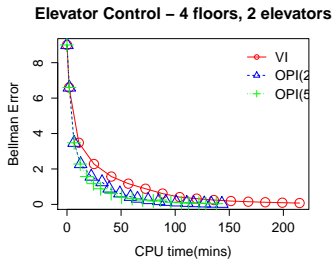

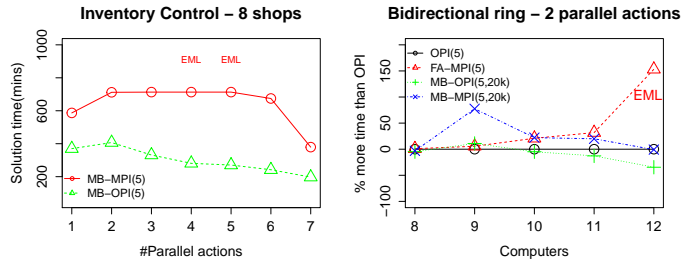

Figure 8: Impact of policy evaluation in Elevators.

Figure 9: Impact of memory bounding. EML denotes Exceeded Memory Limit.

| Domain | # parallel actions | | | | | | # parallel actions | | | | | |
| --- | --- | --- | --- | --- | --- | --- | --- | --- | --- | --- | --- | --- |
| | Compression in V | | | | | | Compression in $\pi$ | | | | | |
| | 2 | 3 | 4 | 5 | 6 | 7 | 2 | 3 | 4 | 5 | 6 | 7 |
| IC(8) | 0.06 | 0.03 | 0.03 | 0.02 | 0.02 | 0.02 | 0.28 | 0.36 | 0.35 | 0.20 | 0.09 | 0.03 |
| Star(11) | 0.67 | 0.58 | 0.50 | 0.40 | 0.37 | 0.35 | $1.8e^{-4}$ | $2.3e^{-4}$ | $2.1e^{-4}$ | $1.9e^{-4}$ | $1.4e^{-4}$ | $9.6e^{-5}$ |
| Biring(10) | 0.96 | 0.96 | 0.95 | 0.94 | 0.88 | 0.80 | $1.1e^{-3}$ | $1.3e^{-3}$ | $1.2e^{-3}$ | $1.1e^{-3}$ | $9.8e^{-4}$ | $7.4e^{-4}$ |
| Uniring(10) | 0.99 | 0.99 | 0.99 | 0.99 | 0.99 | 0.99 | $9.3e^{-4}$ | $1e^{-3}$ | $9.4e^{-4}$ | $8.2e^{-4}$ | $5.2e^{-4}$ | $2.9e^{-4}$ |

Table 1: Ratio of size of ADD function to a table.

the well-established baseline for factored states, SPUDD [1], and factored states and actions FA-MPI(0). Since both are variants of VI we will denote the better of the two as VI in the charts.

**Impact of policy evaluation** : We compare symbolic VI and OPI in Figure 6. For Inventory Control, as the number of parallel actions increases, SPUDD takes increasingly more time but FA-MPI(0) takes increasingly less time, giving VI a bell-shaped profile. An increase in the steps of evaluation in OPI(2) and OPI(5) leads to a significant speedup. For the SysAdmin domain, we tested three different topologies. For all the topologies, as the size of the action space increases, VI takes an increasing amount of time. OPI scales significantly better and does better with more steps of policy evaluation, suggesting that more lookahead is useful in this domain. In the Elevator Control domain (Figure 8) OPI(2) is significantly better than VI and OPI(5) is marginally better than OPI(2). Overall, we see that more evaluation helps, and that OPI is consistently better than VI.

**Impact of pruning** : We compare PI vs. FA-MPI to assess the impact of pruning. Figure 7 shows that with increasing state and action spaces FA-MPI exceeds the memory limit (EML) whereas OPI does not and that when both converge OPI converges much faster. In Inventory Control, FA-MPI exceeds the memory limit on five out of the seven instances, whereas OPI converges in all cases. In SysAdmin, the plot shows the % time FA-MPI takes more than OPI. On the largest problem, FA-MPI exceeds the memory-limit, and is at least 150% slower than OPI. In Elevator control, FA-MPI exceeds the memory limit while OPI does not, and FA-MPI is at least 250% slower.

**Impact of memory-bounding** : Even though memory bounding can mitigate the memory problem in FA-MPI, it can cause a large overhead in time, and can still exceed the limit due to intermediate steps in the exact policy backups. Figure 9 shows the effect of memory bounding. MB-OPI , scales better than either MB-MPI or OPI . In the IC domain, MB-MPI is much worse than MB-OPI in time, and MB-MPI exceeds the memory limit in two instances. In the SysAdmin domain, the figure shows that combined pruning and memory-bounding is better than either one separately. A similar time profile is seen in the elevators domain (results omitted).

**Representation compactness** : The main bottleneck toward scalability beyond our current results is the growth of the value and policy diagrams with problem complexity, which is a function of the suitability of our ADD representation to the problem at hand. To illustrate this, Table 1 shows the compression provided by representing the optimal value functions and policies as ADDs versus tables. We observe orders of magnitude compression for representing policies, which shows that the ADDs are able to capture the rich structure in policies. The compression ratio for value functions is less impressive and surprisingly close to 1 for the Uniring domain. This shows that for these domains ADDs are less effective at capturing the structure of the value function. Possible future directions include better alternative symbolic representations as well as approximations.

# 7 Discussion

This paper presented symbolic variants of MPI that scale to large action spaces and generalize and improve over state-of-the-art algorithms. The insight that the policy can be treated as a *loose* constraint within value iteration steps gives a new interpretation of MPI. Our algorithm OPI computes some policy improvements during policy evaluation and is related to Asynchronous Policy Iteration [9]. Further scalability can be achieved by incorporating approximate value backups (e.g. similar to APRICODD[2]) as weel as potentially more compact representations(e.g. Affine ADDs [3]). Another avenue for scalability is to use initial state information to focus computation. Previous work [13] has studied theoretical properties of such approximations of MPI, but no efficient symbolic version exists. Developing such algorithms is an interesting direction for future work.

## Acknowdgements

This work is supported by NSF under grant numbers IIS-0964705 and IIS-0964457.

## Footnotes

[1] Notice that $T_\pi^Q$ is equivalent to $\pi \times T^Q$ but the former is easier to compute.

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
