[Reviews · NeurIPS 2013]

Submitted by Assigned_Reviewer_4

This paper presents a novel policy iteration algorithm for symbolic
MDPs with factored-action (in addition to factored-state)
dynamics. The algorithm, MB-OPI, yields a way to trade
representational complexity between value and policy iteration for the
class of MDPs defined over algebraic decision diagrams, just as MPI
gives a way to smoothly trade computational complexity. In doing so,
the authors generalize several existing algorithms which consider
factored actions and memory constraints independently.

The main technical challenge is that ADD policy iteration requires
multiplying an explicit policy representation into the current value
function, which can significantly increase its size. The solution is
to control this increase in size by defining a procedure to
conservatively combine policy and value diagrams using a pruning
procedure, rather than naively multiplying them. Results are
presented in terms of solution time, and show a ~2-6x improvement over
existing approaches.

This paper is technically sound and well written. The authors make a
theoretical contribution to the literature on symbolic MDP planning by
introducing the concept of pruning as an alternative to ADD products,
and proving that this satisfies the guarantees of MPI. Also couched
as generalization to existing work in symbolic dynamic programming,
and appears to be state of the art for planning with factored actions

Empirical results support the idea that pruning offers an MPI approach
to SDP planning that avoids representational bloat, and offers a
several factor speed up

The paper is also generally well written and easy to follow.

If possible, I would suggest adding more background on SDP solving
using ADDs for representing value and DBNs, the basic policy iteration
approach using ADD product, and the difference between multiplying pi
into V vs. pruning V with pi.

It would also be nice to have (a) discussion of practical problems
with many parallel actions for which factoring actions is critical,
and (b) a toy test case with large parallel actions that highlights
the best-case improvement over SPUDD and FAR.


Some notes on clarity that might be helpful:

053, "enforcement of policy constraint": 'constraint' hasn't been defined yet, and only makes sense if you remember to view policy iteration as policy-constrained value iteration

060, Figure 1: ordering of state and action nodes would be more readable if they were interleaved or stacked (something consistent)

060, Figure 1: as state variables these are propositions, not predicates, so might be better to use underscores (e.g. reboot_c1)

095, "marginalization operators": examples include marginalization but also max. should reword for correctness

110, equation 1: this was confusing without referring back to SPUDD paper. I suggest adding 2 things: (a) explanation of how expectation for factored models turns into a product of DBNs, and that sums can be pushed in, and (b) simple explanation that "primed" literally adds a prime to each state variable, and is necessary to make the ADD operations well defined (saying "swaps the state variables X in the diagram V with next state variables X′" can be confused with more complicated mapping issues)

152, policy ADD: Would be helpful to have a sentence like "Intuitively, this representation makes it possible to express 1-step policy backup as the ADD product of a policy with a value function".

179, Figure 4: caption uses C to denote policy, but figures use \pi. other places too

206, "size of a Bellman backup": usually think of backups in terms of computational complexity, so should clarify that "size" refers to the data structure that results from applying eq1. also would be helpful to intuitively explain what this looks like, as opposed to pi (both have action variables, so why is pi generally bigger??)

206, (due to…): for clarity, would help to clarify that pi is bigger because it represents joint-actions, whereas backup only represents value and (product of) factored actions

212, "only those paths in D that": add "in D" to make clear that policy is being applied to value function. otherwise can be confused with original definition of policy

247, eq3: two extra close parens

252, "sandwiched": intuitive, but perhaps too informal (though who am I to say such a thing?)

278, "parameterized by k, …": missing comma

327, Figure 6: colors for OPI 2 and 5 are reversed in b... I think.
Summary: This paper presents a well-defined improvement to decision-diagram
based planning in symbolic MDPs. Empirical and theoretical results
suggest that their algorithm is the state of the art for planning with
factored actions.

Submitted by Assigned_Reviewer_5

This paper introduces an improvement to symbolic policy iteration for domains with factored actions. The core idea seems to be that we can take some liberties with the symbolic backup operations to reduce the size of the resulting ADDs, and that the particular way that this is done is by performing a more general backup (rather than an on-policy backup) for some actions, when doing so does not increase the resulting expression. This is proven to converge, and some thorough and reasonable impressive experimental results are given, though I do not know enough about symbolic policy iteration to determine whether or not they are exhaustive.

Both symbolic approaches and factored actions are interesting and under-explored, so I am positively disposed toward the paper.
The main difficulty I had was that it is not explained how \pi is represented as an ADD, so that \pi is introduced as a constraint in section 3. Some more explanatory material here - perhaps giving an example of an on-policy vs. off-policy action backup links to the result of the pruning, would really help. As it is the reader has to piece together what the pruning operator does from some math and some diagrams, before its high-level explanation-which as far as I can understand is actually quite simple - is given in passing. This is made extra confusing in Figure 4, when D and C presumably mean D and \pi.

Unfortunately this, combined with only passing familiarity with symbolic approaches, made the paper quite hard to understand, when it probably should not be.

Otherwise I only have small writing comments:
o "flat actions" might better be described as "atomic actions". "Flat" is often the opposite of hierarchical.
o "assume a small flat action space" -> "assumes a small flat ..."
o "Factored State and Actions Spaces" -> "Factored State and Action Spaces"
o A parenthetical citation is not a noun.
o assignments OF n boolean variables to A real value
o "interestingly, the first approach to symbolic planning in MDPs, was a version" (delete comma)
o The graphs are all tiny. I appreciate the space constraint, but it looks like you can eliminate some whitespace.
o Please use full citation information and properly format your references (including capitalization).
Summary: An interesting paper that presents a technical improvement to symbolic backups that improve their performance. Often difficult to understand.

Submitted by Assigned_Reviewer_6

This paper describes a pruning technique that enables symbolic modified policy
iteration in large factored action MDPs. Their technique, like regular MPI,
generalizes policy iteration and valuation, incorporates prior (orthogonal)
work on partially bound actions, and is experimentally validated.

I weakly recommend this paper for acceptance. The main contribution appears to
be a non-trivial implementation detail, but their experiments show that (a)
pruning by itself is helpful for value iteration, (b) pruning is required for
modified policy iteration, which is often not possible for memory reasons, and
(c) that modified policy iteration improves convergence in factored action
MDPs.

The paper is well motivated, but the notation is inconsistent in places and
often hard to follow. e.g., Alg 3.2 is called Prune, but it is used as \cal P
elsewhere, it is not obvious from the text that T^Q(V) is a function of
states and actions, or even that the variables are binary.

My main concern with the paper is that I could not follow the details enough to
completely understand the statement of theorem 1. In particular, it is not
clear why \hat T^Q_\pi can be different than T^Q_\pi. Is it necessary to prune
at every step, or is it sufficient to prune only once? Is it the repeated
pruning that causes the overestimation? or is the convergence theorem the same
for FA-MPI and OPI?

Proposition 2 seems trivial. Is there any guarantee on how much smaller the
pruned tree will be?

Summary: I recommend that this paper be accepted. From a high level it is well motivated and clearly written, and the experiments demonstrate its ability to tackle previously intractable problems.
Author Feedback

Author rebuttal: Thanks to the reviewers for their comments. The reviewers seem to
agree with the technical merits of the paper and mainly raised questions
about the writing/clarity. We address the main comments of the reviewers
here. All the other minor comments will be addressed in the final camera
ready version.

Reviewer 1:

Re: "The main difficulty I had was that it is not explained how \pi
is represented as an ADD, ...."

See paragraph 2 of Section 3: ``We represent the policy using a
Binary Decision Diagram (BDD) with state and action variables where
a leaf value of $1$ denotes a combination of action variables that
is the policy action, and a leaf value of $-\infty$ indicates
otherwise.''. Figures 4 and Figure 5 also give examples of
policies.

Re: "Some more explanatory material here - perhaps giving an
example of an on-policy vs. off-policy action backup links to the
result of the pruning, would really help."


This is shown in Figure 5, where the figures in the extreme right
show policy backup and off-policy backup(a single node of value
1.5). We will work to improve the clarity by rewording captions and
referencing text and rearranging the diagrams.

Reviewer 2:

Re: "As it is the reader has to piece together what the pruning
operator does from some math and some diagrams, before its
high-level explanation-which as far as I can understand is actually
quite simple - is given in passing."

We will include some of the higher-level remarks earlier in the
revised paper, which will look something like,
this paper views policy iteration as a ``loosely''
constrained value iteration, where the resulting backup operator
is more efficient than both a policy and a bellman
backup. In terms of flat MDP states, our backup computes a policy
improvement in some states and exact policy backup in other
states. This trade-off is done dynamically, and exploits the
factorization of actions and policy.

Reviewer 3:

" ... it is not clear why \hat T^Q_\pi can be different than
T^Q_\pi."

This is because the policy (constraint) is not strictly enforced in
\hat T^Q_\pi compared to what is required by T^Q_\pi. When the policy
constraint is not enforced, the backup maximizes over all actions,
which leads to overestimation.

This was shown in Figure 5, in the rightmost diagrams, where \hat
T^Q_\pi (named P(T^Q)) and T^Q_\pi (named T^*) are different. We
will standardize the notation between text and figure in the
revised paper.

"Is it necessary to prune at every step, or is it sufficient to
prune only once? Is it the repeated pruning that causes the
overestimation?"

Pruning is for efficiency, so while not necessary, it is best done
at every iteration.

Any amount of pruning can cause overestimation. Note that the
nature of the overestimation is such that we still converge to
optimal policies.

"or is the convergence theorem the same for FA-MPI and OPI?"

Both FA-MPI and OPI converge to the optimal policy, but via a
different sequence of intermediate value functions/policies. FA-MPI
mimics MPI exactly, but with the factored action representation.
OPI computes more efficient overestimations of policy backups, but
never overestimates a Bellman backup. OPI converges because its
value is sandwitched between the policy backup and the Bellman
back, both of which converge. That is the significance of Theorem
1, described in the paragraph below Theorem 1 on Page 5.

"Proposition 2 seems trivial. Is there any guarantee on how much smaller the pruned tree will be?"

Proposition 2 is not difficult but important. It guarantees that
the result of pruning never increases the size of the BDD, while
without pruning the size could grow geometrically. The actual
reduction of size is domain-dependent.

In general, due to space constraints we were unable to add more background
material on SDP. The authors also acknowledge a slight inconsistency of
notation between the text and figures, which will be corrected in the camera
ready version.